# Electrochemical Immunodetection of *Bacillus anthracis* Spores

**DOI:** 10.3390/s25195948

**Published:** 2025-09-24

**Authors:** Karolina Morawska, Tomasz Sikora, Aleksandra Nakonieczna, Renata Tyśkiewicz, Monika Wiśnik-Sawka, Łukasz Osuchowski, Paulina Osuchowska, Michał Grabka, Zygfryd Witkiewicz

**Affiliations:** 1Department of Dosimetry and Contamination Detection Systems, Military Institute of Chemistry and Radiometry, Al. Gen. Antoniego Chruściela “Montera” 105, 00-910 Warsaw, Poland; k.morawska@wichir.waw.pl (K.M.); m.wisnik-sawka@wichir.waw.pl (M.W.-S.); 2Biological Threats Identification and Countermeasure Center, Military Institute of Hygiene and Epidemiology, Lubelska 4, 24-100 Puławy, Poland; aleksandra.nakonieczna@wihe.pl; 3Analytical Laboratory, Łukasiewicz Research Network—New Chemical Syntheses Institute, Al. Tysiąclecia Państwa Polskiego 13A, 24-110 Puławy, Poland; renata.tyskiewicz@ins.lukasiewicz.gov.pl; 4Institute of Optoelectronics, Military University of Technology, ul. gen. Sylwestra Kaliskiego 2, 00–908 Warsaw, Poland; lukasz.osuchowski@wat.edu.pl (Ł.O.); paulina.osuchowska@wat.edu.pl (P.O.); 5Institute of Chemistry, Military University of Technology, ul. gen. Sylwestra Kaliskiego 2, 00-908 Warsaw, Poland; michal.grabka@wat.edu.pl (M.G.); zygfryd.witkiewicz@wat.edu.pl (Z.W.)

**Keywords:** electrochemical biosensor, *Bacillus anthracis* spore detection, microbiological analysis, portable sensor, immunoelectrochemical detection

## Abstract

**Highlights:**

**What are the main findings?**
Electrochemical biosensor detects *Bacillus anthracis* in minutes with high selectivity.Novel thiol-modified gold electrodes enable sensitive and specific pathogen detection.Limit of detection reaches 10^3^ CFU/mL, ensuring early anthrax spore identification.

**What is the implication of the main finding?**
This portable, low-cost biosensor offers an alternative to PCR and ELISA in field settings.The method requires minimal sample preparation and is suitable for on-site diagnostics.

**Abstract:**

The Centers for Disease Control and Prevention (CDC) classifies *Bacillus anthracis* as one of the most dangerous pathogens that may affect public health and national security. Due to its importance as a potential biological weapon, this bacteria has been classified in the highest category A, together with such pathogens as variola virus or botulinum neurotoxin. Characteristic features of this pathogen that increase its military importance are the ease of its cultivation, transport, and storage and its ability to create survival forms that are extremely resistant to environmental conditions. However, beyond bioterrorism, *B. anthracis* is also a naturally occurring pathogen. Anthrax outbreaks occur in livestock and wildlife, particularly in spore-contaminated regions of Africa, Asia, and North America. Spores persist for decades, leading to recurrent infections and zoonotic transmission through direct contact, inhalation, or consumption of contaminated meat. This work presents a new electrochemical method for detecting and quantifying *B. anthracis* in spore form using a selective immune reaction. The developed method is based on the thiol-modified electrodes that constitute the sensing element of the electrochemical system. Tests with the *B. anthracis* spore suspension showed that the detection limit for this pathogen is as low as 10^3^ CFU/mL. Furthermore, it was possible to quantify the analyte with a sensitivity of 11 mV/log (CFU/mL). Due to several features, such as low unit cost, portability, and minimal apparatus demands, this method can be easily implemented in field analyzers for this pathogen and provides an alternative to currently used techniques and devices.

## 1. Introduction

Bacteria, viruses, and other microorganisms are common in the natural environment. Most of them perform essential environmentally beneficial activities, but certain potentially harmful microorganisms can have profound effect on humans and animals and may cause a variety of infectious diseases [1]. From a military perspective, there are many pathogenic bacteria that could be considered possible biological warfare agents [2,3,4,5]. These bacteria are resistant to weather conditions and have the ability to cause severe diseases with a high risk of lethal effects, and most of the human population is susceptible to them. Furthermore, many lethal microorganisms can grow in the environment and remain viable for several years [6,7]. Effective bacterial testing requires analysis methods that meet a number of challenging factors. The most important limitations to the usefulness of microbiological testing are the analysis time and test sensitivity [8]. Methods for detecting bacteria must be rapid and very sensitive because the presence of even a single pathogenic organism in the body or food can already constitute an infectious dose [9,10]. Traditional methods for bacteria detection include standard clinical microbiological tests. Currently, most microbiology research is centralized in large stationary laboratories as it requires complex instrumentation and highly skilled technical staff [11]. Identification with the use of phenotypic methods is a long-term process due to approximately 24 to 48 h of incubation required for the cultivation of a pathogen present in a clinical sample and another 24 or more hours for its accurate identification [12,13,14,15,16]. Reducing the time necessary to identify a microorganism is possible by using modern and improved technological processes already available in routine microbiological diagnostics. Therefore, innovative detection methods are being sought, which undoubtedly include biosensors: devices that transform the biological signal resulting from the reaction of a biological molecule with the determined compound into a measurable analytical signal [17]. Biosensors are considered promising tools for rapid detection and identification of bacteria [18]. The fundamental part of a biosensor is its detection layer, which is a system that recognizes the analyte being determined and is able to react with it [19,20,21,22]. The elements that recognize the marked substance are biological molecules such as enzymes, nucleic acids, and antibodies [23]. These molecules are most often immobilized on the surface of the detection part of the device. The type of detection layer used depends on many factors, primarily the nature of the substance being determined, its concentration and volume, and whether the measurement is one-time or continuous. A characteristic feature of biomolecules is their high selectivity for the substrate with which they react.

Electrochemical biosensors are a relatively new type of sensors used for detecting biomolecules such as DNA or proteins [24,25,26,27]. The combination of rapid, sensitive, selective, and cost-effective molecule detection using electrochemical methods and areas such as genetics, biochemistry, and molecular biology expands the potential applications of biosensor-based devices in medicine [28,29,30]. These devices are also valuable due to their potential for miniaturization, ease of use, and low equipment costs [31]. The basic principle of electrochemical sensors involves the electroactive oxidation or reduction of a compound on the surface of the working electrode, which is subjected to a predetermined potential, and the change in electrical parameters resulting from the redox reaction, dependent on the type or concentration of the analyte. Electrochemical detection is performed using a three-electrode system: a working electrode where the electron transfer reaction occurs, a reference electrode that maintains a stable potential relative to the working electrode, and an auxiliary electrode that reduces the solution resistance, among other functions.

In recent years, considerable attention has been given to developing rapid, low-cost, and field-deployable biosensors for the detection of *B. anthracis*, due to the limitations of traditional methods such as PCR (requiring complex sample preparation and trained personnel), ELISA (involving long incubation times), and bacterial culture (time-consuming and potentially hazardous). The electrochemical immunosensor described in this study introduces a simplified detection strategy using a gold electrode functionalized with thiol-based SAM and gold nanoparticles as antibody carriers, enhancing sensitivity through increased surface area for antigen binding.

What distinguishes our approach is the integration of a regenerable gold surface, a straightforward electrochemical readout based on cyclic voltammetry, and the potential for miniaturization into portable formats suitable for rapid, on-site diagnostics. Unlike some existing biosensors that rely on aptamers or enzymatic signal amplification, our system is based on direct antigen–antibody interaction with minimal reagent complexity. This platform offers a promising alternative for early detection of anthrax spores in environmental or emergency settings, especially where rapid decision-making is critical.

This work presents a method for manufacturing *B. anthracis*-selective electrodes constituting a sensor element of an electrochemical system. The electrodes were prepared using a commercially available screen-printing transducer in a three-electrode system. The surface modification of the electrode included a number of processes, such as cleaning, immobilization of the thiol monolayer, immobilization of the gold nanoparticle monolayer, and covalent immobilization of the antibody layer through oxidation of disulfide bridges. Testing the sensor elements prepared in this manner against the target analyte at various concentrations and in the presence of other pathogens in samples enabled the determination of the most important parameters of the new analytical method, including detection limit, sensitivity, response time, and selectivity. In addition, conclusions were drawn on the possibility of implementing the method in portable analyzers, which would enable the detection of this analyte in field conditions.

## 2. Materials and Methods

### 2.1. Materials

In this study, the following chemical and biological materials have been used for biosensor preparation and testing:-MilliporeSigma (Burlington, MA, USA): Bovine serum albumin (BSA), gold colloidal nanoparticles (nanoparticles are spherical in shape with an average diameter of approximately 20 nm), potassium ferro- and ferricyanides, phosphate-buffered saline (PBS—mM NaCl, 2.7 mM KCl, 1.8 mM Na_2_HPO_4_, 10 mM KH_2_PO_4_), 4,4′-thiobisbenzenethiol (TBBT), and tris(2-carboxyethyl)phosphine hydrochloride (TECP);-Nitrogen gas (N_2_ 5.0, Messer Polska Sp. z o.o., Poland);-Avantor Performance Materials Poland S.A. (Gliwice, Poland): Sulphuric acid, potassium hydroxide, ethanol and methanol;-Thermo Fisher Scientific (Waltham, MA, USA): 5.1 mg/mL B. anthracis Spore Antigen Monoclonal Antibody (SA26), Goat Anti-Mouse IgG (H + L) Secondary Antibody, and six types of monoclonal antibodies against *B. anthracis* spores (IgG2a class antibody clones: B57G, G46D, SA27, 3G302, and 5E218 in PBS pH 7.2 solution);

The antibodies used in this study are designed to recognize surface antigens present on *B. anthracis* spores. Although the exact epitopes are not fully characterized, available data indicate that monoclonal antibody binds to a spore-associated antigen. These molecular markers are known to be characteristic of the *B. anthracis* exosporium and play a role in the specificity of immunodetection.

-Bacterial suspensions prepared and delivered by the Biological Threats Identification and Countermeasure Center of the Military Institute of Hygiene and Epidemiology (BTICC, Puławy, Poland): Spore suspensions (*Bacillus mycoides* 21929, *Bacillus thuringiensis* ATCC 35646, *B. thuringiensis* ATCC 33679, *B. thuringiensis* ATCC 10792 T, *Bacillus cereus* ATCC 10876, *B. cereus* ATCC 13472, *B. cereus* ATCC 14579 T, *Bacillus subtilis* ATCC 6633, *B. anthracis* 34F2), and vegetative cell suspension of *Escherichia coli*.

All aqueous solutions were prepared with deionized water that was highly purified using a Milli-Q reagent-grade water system (Millipore, Burlington, MA, USA).

### 2.2. Methods and Instrumentation

Gold electrodes from Metrohm (Herisau, Switzerland), model 220AT, were used as an electrochemical transducer and subjected to surface modification as part of this work.

The cyclic voltammetry (CV) method was used to modify the electrode surfaces and determine the analyte. The research used the SP-200 potentiostat with Ec-Lab software (Bio-Logic, Seyssinet, Paris, France) in a three-electrode system.

To perform an analytical measurement, the following steps were required:

Performing a measurement in PBS buffer, pH = 7.4, with 1 mM ferrocyanide added. This served as a reference measurement. Rinsing and drying the electrode. Placing the electrode in the test solution (containing *B. anthracis* spores) and incubating for approximately 5 min. Rinsing and drying the electrode. Performing another measurement in PBS buffer, pH = 7.4, with 1 mM ferrocyanide added. The entire procedure takes approximately 15 min.

Due to the availability of many antibodies specific to *B. anthracis* on the market, the research team from BTICC in Puławy conducted an analysis of 6 antibodies directed against *B. anthracis* to select the most advantageous one. Antibodies were tested against standardized suspensions: vegetative cells of the *E. coli* strain, spore cells of *B. anthracis*, and spore cells of 8 other strains of the *Bacillus* genus. Percentages of spore-binding activity of the individual tested antibodies were used for Principal Component Analysis (PCA) preparation. The PCA was performed using Statistica 13.3 (StatSoft. Inc., Cracow, Poland).

Microscopic characterization of the modified electrode surfaces was performed using a Laser Scanning Confocal Microscope (LSM 700, Zeiss Axio Observer.Z1, Gottingen, Germany) equipped with an expanded illumination system operating at a wavelength of 405 nm. Additionally, high-resolution imaging was carried out using a Scanning Transmission Electron Microscope (STEM) (Quanta FEG250, FEI, Hillsboro, OR, USA) to visualize the morphology and structure of the electrode surfaces at the nanoscale.

### 2.3. Surface Modification of Gold Electrodes

The main objective of the conducted research was to modify a commercially available gold electrode, creating an analytically active layer and enabling the electrode’s use for *B. anthracis* detection. The modification was designed to include the formation of a dithiol layer and gold nanoparticles on the surface of the gold electrode, followed by the directional immobilization of primary antibodies through the reduction of disulfide bonds and the high affinity of gold–sulfur. In this method, antigens present on the surface of *B. anthracis* spores are directly bound to the antibodies. The analytical signal is generated through the redox reaction of the [Fe(CN)_6_)]^3−/4−^ marker. The absence of an electrolytic medium in the measurement system would lead to a significant reduction in ionic conductivity between the electrodes, effectively preventing current flow and the detection of any electrochemical signal. The electrolyte plays a crucial role in facilitating charge transport in solution, enabling electron exchange between the electrode and the analyte. Therefore, its presence is essential for the proper functioning of the biosensor, ensuring signal stability and measurement reliability.

The developed modification of the working electrode comprised five stages:Electrochemical cleaning of the electrode;Formation of the TBBT (4,4′-thiobisbenzenethiol) layer;Immobilization of colloidal gold nanoparticles;Immobilization of antibodies;Blocking free spaces on the electrode with albumin molecules (BSA: bovine serum albumin).

The electrode and the successive stages of modification are shown in Figure 1.

#### 2.3.1. Electrochemical Cleaning of the Working Electrode Surface

The first step was the electrochemical cleaning of the electrode surface to prepare it for modification. Initially, the electrode was thoroughly rinsed with ethanol and ultrapure, deionized water. After drying the surface with gaseous nitrogen, 50 µL of 0.5 M sulfuric acid solution was deposited. Using a potentiostat, the current intensity was measured while sweeping the potential from 200 mV to 1100 mV at a scanning rate of 100 mV/s.

Further modifications were carried out solely on the surface of the working electrode. To facilitate the application of the TBBT solution directly onto the working electrode and to separate the RE and CE, the design team from the Military Institute of Chemistry and Radiometry developed a dedicated electrode holder (Appendix A). The holder has an upper opening with the same diameter as the WE. The electrode is placed in the groove of the holder and tightly sealed during the immobilization of subsequent layers. To minimize solution evaporation and maintain uniform conditions, the opening of the holder was sealed. This holder was engineered to isolate the working electrode, minimizing cross-contamination and evaporation, ensuring stable immobilization of functional layers.

#### 2.3.2. Deposition of the TBBT Layer

After cleaning the electrode surface, the second step of the modification commenced: the immobilization of 4,4′-thiobisbenzenethiol. The TBBT thiol monolayer was used in this work due to its stability and other desirable properties important in electrochemical measurements (such as low electron transfer resistance) [32]. Thiols (compounds containing a thiol group -SH) were used for electrode modification to enhance the adsorption of proteins, enzymes, and other biomolecules on the electrode surface. They have the ability to form self-assembling monolayers on gold surfaces. In this step, the sulfur reacts with gold to form a covalent bond, followed by the self-assembly of the resulting layer [33].

This specific function of thiols is only useful on the working electrode, which is the site where electrochemical processes related to the sensor’s investigated reactions occur. Applying thiols only to the working electrode helps to avoid undesired chemical reactions on the reference electrodes, which could disrupt measurements or interfere with electrochemical processes.

The total length of the TBBT molecule—from the gold surface to the end of the tert-butyl group—is approximately 0.9 nm. This value is consistent with previous studies on the structure and properties of thiol-based self-assembled monolayers, where similar structures exhibit lengths ranging from 0.7 to 1.1 nm, depending on the degree of molecular ordering and the deposition conditions.

The electrode was placed in the holder groove and tightly sealed. After that, 20 μL of 1 mM TBBT solution in ethanol was dropped through the opening window. Thiol immobilization was conducted at room temperature for 30 min. After incubation, the electrode was thoroughly rinsed with ethanol and ultrapure deionized water and dried with gaseous nitrogen.

#### 2.3.3. Restoration of the Nano-Gold Layer

In the next step, it was necessary to regenerate the gold surface because thiol modification can lead to chemical changes on the surface and decrease chemical activity. Although the base electrode is indeed made of gold, the incorporation of gold nanoparticles plays a role in enhancing the overall performance of the biosensor. Specifically, the AuNPs serve to significantly increase the electroactive surface area, allowing for a higher density of antibody immobilization compared to a flat gold surface. This directly contributes to improved sensitivity of the detection system. Additionally, restoring the gold surface improves the electrode’s conductivity, which is beneficial for the transduction of electrical signals associated with the chemical reaction and ensures the stability of the modifying layer. The electrode was placed in the holder, and then, 10 µL of gold nanoparticle solution was applied, and the holder’s opening window was sealed. Immobilization occurred at 8 °C and lasted for 2 h. After the appropriate time, the electrode was thoroughly rinsed with ethanol and ultrapure deionized water and dried with gaseous nitrogen.

#### 2.3.4. Immobilization of Primary Antibodies

The fourth step in the electrode modification process involved the immobilization of primary monoclonal antibodies specific to *Bacillus anthracis* onto the restored gold surface of the working electrode. To facilitate efficient binding to gold, the antibody solution was pretreated to reduce disulfide bonds and generate free thiol groups. For this purpose, a 10 mM solution of tris(2-carboxyethyl)phosphine hydrochloride in phosphate-buffered saline (PBS, pH 7.4) was prepared. Then, 2 µL of the TECP solution was added to 200 µL of antibody solution (20 ng/mL), and the mixture was incubated for 1 h at room temperature.

Following this activation step, 20 µL of the treated antibody solution was applied to the WE, which had been previously rinsed with 0.1 M PBS. The electrode was placed into the dedicated holder and incubated for 2 h at room temperature. After incubation, the electrode was thoroughly rinsed with PBS and dried under a stream of nitrogen gas.

To minimize nonspecific interactions, the electrode surface was subsequently passivated with a 1% (*w*/*v*) solution of bovine serum albumin (BSA) in PBS. This step effectively blocked any remaining free binding sites and prevented unintended protein adsorption, which could otherwise result in false-positive or false-negative responses.

Moreover, steric hindrance was reduced by using an optimized BSA concentration and controlled incubation conditions. Since BSA was applied only after antibody immobilization, it did not interfere with the availability of the antibody binding sites. Additionally, the presence of gold nanoparticles as antibody carriers provided spatial separation from the electrode surface, further limiting steric interference and enhancing antigen accessibility.

### 2.4. Selection of a Specific Antibody

The selection of antibodies for the sensors was based on the results of specificity tests. These tests were conducted on a group of commercially available antibodies selective for *B. anthracis* using standardized solutions of bacteria from the *Bacillus* genus (including *B. anthracis*) and of *Escherichia coli*. In the tests, different types of antibodies were immobilized on polystyrene substrates (ELISA 96-well plates) and then incubated with standardized bacterial solutions. After the incubation period, the bacterial solutions were quantitatively transferred onto a Petri dish, where unbound bacteria were counted using an optical microscope. The same procedure was followed in the negative control, but the solutions were incubated on substrates without immobilized antibodies. The difference in the number of counted bacteria between the antibody-coated sample and the negative control determined the test result. The measurement results, expressed as the percentage of bound antigens, are shown in Table 1.

The percentage of antigen-binding values presented in Table 1 for all antibodies is significantly higher for *B. anthracis* than for the other antigens in the test set. To illustrate the specificity of antibodies and the differences between them, the principal component analysis (PCA) was performed. The first two principal components explained 97.14% of the total variance, with principal component 1 (PC1) accounting for 91.10% and PC2 accounting for 6.04%. A PCA biplot (Figure 2) presents the principal component scores of the samples (triangles: *Bacillus* strains; red dot: *E. coli*) and the loadings of the variables (vectors: antibodies).

Antibodies exhibit some differences in their specificity toward the entire group of antigens, as clearly illustrated by the antibody vectors in Figure 2. Additionally, an analysis of the scoring chart indicates that the *B. anthracis* strain clearly differed from the others in terms of affinity for antibodies. The remaining bacteria (*B. thuringiensis*, *B. cereus*, *B. mycoides*, *B. subtilis*, and *E. coli*) in Figure 2 form a fairly compact group (blue loop), far removed from *B. anthracis*.

To determine the types of antibodies with the highest selectivity toward *B. anthracis*, their selectivity coefficients (S) were calculated and compared. The coefficients were calculated based on the values from Table 1 using the following equation (Equation (1)):(1)$$S=\frac{{Resp.}_{B.anthracis}}{\sum {Resp.}_{non-selective}}\cdot 100\%$$
where

${Resp.}_{B.anthracis}$—response to *B. anthracis*;

$\sum {Resp.}_{non-selective}$—sum of responses to all bacteria from the test set, except *B. anthracis*.

The values of selectivity coefficients for individual antibodies are presented in Table 2.

Although selectivity coefficients around 38–47% may appear modest, they reflect relative affinity in comparison to control strains. These values are consistent with early-stage biosensor development and are sufficient for distinguishing *B. anthracis* against closely related *Bacillus* spp.

Analysis of the data in Table 2 shows that the values of the selectivity coefficients of all antibodies are similar, and all of them can be described as highly selective for this antigen. On this basis, it can be concluded that all tested antibodies should be suitable for the selective detection of *B. anthracis*. The selection criteria should include other factors, such as the availability of a given antibody, stability over time, etc. Based on this criterion, the SA27 antibody was selected for further testing.

## 3. Results

### 3.1. Cyclic Voltammetry (CV) Measurements for Monitoring the Deposition Processes of Successive LLayers on the Surface of the Screen-Printed Gold Electrode

The effectiveness of processes such as electrode cleaning, the deposition of self-assembled monolayers, and their interactions with selected analytes were investigated using CV. Conducting measurements through electrochemical techniques requires the presence of a compound in the solution undergoing oxidation and reduction reactions. In cases where the analyte is not electroactive, redox probes such as [Fe(CN)_6_]^3−/4−^ are employed, which were chosen for measurements in this study. A 1 mM solution of K_3_[Fe(CN)_6_]/K_4_[Fe(CN)_6_] in 0.1 M phosphate-buffered saline was utilized. CV measurements were conducted over a potential range of −200 mV to 600 mV at a scan rate of 100 mV/s. Changes in the signal for successive modification steps are shown in Figure 3a–d.

As a parameter determining surface cleanliness, the charge on the electrode was adopted. During the cyclic voltammetry measurement in the cleaning stage, inhibition of peak growth and stabilization of the current signal were observed upon reaching a charge value of approximately 20 C. After obtaining such a signal, the charge remained unchanged, and the signal was repeatable, indicating the absence of any contaminants on the electrode surface that could interfere with the thiol deposition process (Figure 3a). Clear shifts in oxidation and reduction peaks, as well as changes in their heights depending on the layer deposited on the electrode surface, are visible, indicating the successful immobilization of layers. The results of CV measurement for successive electrodes, after immobilization of primary antibodies for a solution of 1 mM K_3_[Fe(CN)_6_]/K_4_[Fe(CN)_6_] in 0.1 M PBS, indicate the high repeatability of the immobilized antibody layers.

### 3.2. Investigation of the Modified Gold Electrode After Incubation with B. anthracis Spores

The results for the electrodes after incubation with *B. anthracis* at a concentration of 10^8^ CFU/mL (colony-forming units; a bacterial suspension in 0.1 M PBS, pH 7.4) are presented in Figure 4. The graph presents the recorded signals obtained for successive, identically modified gold electrodes. The recorded voltametric curves demonstrate the high repeatability of surface modification. Each of the presented curves was recorded for a separate biosensor for an identical concentration of the spore suspension.

Upon bacterial capture by antibodies, there is a potential shift and a decrease in current: the average potential value is 125 +/− 2.8 mV, and the average current value is 24.82 +/− 0.79 µA. The change in the spectrum profile, the shift in the maximum oxidation and reduction peaks after applying *B. anthracis* spores, indicates that the adopted method allows for effective detection of their presence.

The next step was to verify the discrimination capabilities of the modified electrode in terms of the differentiation of bacterial concentrations. Solutions of spores at concentrations of 10^3^, 10^4^, 10^5^, 10^6^, 10^7^, and 10^8^ CFU/mL were applied (a bacterial suspension in 0.1 M PBS at pH 7.4). The results are presented in Figure 5 and Figure 6. High repeatability of detection was demonstrated across the entire concentration range, which confirms the correctness of the entire procedural process and the validity of all electrode modifications, enabling the analysis of the subject of the project: *B. anthracis* spores.

As shown in Figure 6, as the concentration of *B. anthracis* in the sample increases, the maximum reduction peak shifts toward higher potentials, and the corresponding current decreases. These two values will form the basis for qualitative analysis.

The conducted measurements are quasi-quantitative in nature. This is not a direct measurement and requires prior measurement in a buffer solution, followed by measurement in the analyzed solution. The following formula was used for calculations (Equation (2)):(2)$$\frac{I_{n}-\;I_{0}}{I_{0}}\times 100\%$$
where

I_n_—the current intensity measured for the sample after incubation;

I_0_—the current intensity before incubation.

An analogous formula was used for potentials. The reference point for the analysis was the baseline measurement, i.e., the measurement conducted in a solution without *B. anthracis* spores. For over one hundred measurements, the average value of the blank signal was determined as I_0_ = 37.15 μA (for currents) and E_0_ = 63.80 mV (for potentials). The standard deviations SD_I_ = 0.76 μA and SD_E_ = 1.37 mV, respectively.

Subsequently, a series of measurements was conducted for samples containing *B. anthracis* spores in the concentration range of 10^3^–10^8^ CFU/mL. For the lowest tested concentration (10^3^ CFU/mL), the average current and potential values at the maximum reduction peak were I_0_ = 35.9 μA and E_0_ = 67.9 mV, respectively.

### 3.3. Examination of the Layer of Antibodies Immobilized on the Electrode Surface

Microscopic verification was conducted to demonstrate the proper antibody immobilization on the electrode surface. Two methods were applied for visualizing the electrode surface. The first visualization was made using a Laser Scanning Confocal Microscope LSM 700 Zeiss Axio Observer.Z1 (CSLM) with an expanded illumination system at a wavelength of 405 nm. Also, images of the electrodes were taken with a Scanning Transmission Electron Microscope (STEM) (Quanta FEG250, FEI). The surface of each electrode was observed in high-vacuum mode using an Everhart–Thornley detector (ETD). The results of the examination are depicted in Figure 6.

The examination was performed on two types of electrodes (Figure 7) The first electrode was clean and uncoated with antibodies. The second electrode was coated with the specific primary antibodies and subsequently coated with secondary antibodies A-31561. Primary antibodies were immobilized on the electrode surface as part of the functionalization process. To enhance their visibility under the microscope, secondary antibodies were applied. These secondary antibodies specifically bound only to the regions on the electrode where the primary antibodies were already present. In areas without immobilized primary antibodies, no binding of secondary antibodies occurred. This approach serves to confirm the successful immobilization of primary antibodies on the electrode. If the immobilization process had failed, the images obtained before and after the application of secondary antibodies would appear identical.

Figure 7a shows the visualization of the electrode morphology of the uncoated electrode and Figure 7b of the coated electrode. Both electrodes were observed under a CSLM. The red color on the fluorescent image from the CSLM examination indicates the integrity of the active antibodies with the electrode surface (Figure 7d). Therefore, the average fluorescence intensity was obtained for the areas verified on the electrodes: 0.34 and 4.73. The histogram effectively displayed the fluorescence intensity of the coated electrode (Figure 7d) compared to the uncoated electrode (Figure 7c). Furthermore, the morphology of the electrode surface was visualized using the STEM (Figure 7e,f). In Figure 7e, the raw porous surface of the electrode is shown. Figure 7f presents the surface covered (sealed) by the primary and secondary antibodies. The specific connection strongly depends on the properties of the substrate material.

## 4. Discussion

This study presents an electrochemical immunosensor designed for the detection of *Bacillus anthracis* spores. The developed system features a simple design, a short analysis time, and potential for on-site application, which distinguishes it from traditional diagnostic techniques such as PCR and ELISA.

The use of a gold nanoparticle (AuNP)-enriched electrode surface allowed for increased active surface area and thus a higher density of antibody binding sites, significantly improving the sensor’s sensitivity. The choice of a gold electrode and a SAM layer based on 4-tert-butylbenzenethiol enabled stable and effective immobilization of antibodies specific to *B. anthracis* exosporium antigens. The application of a BSA-based blocking strategy minimized nonspecific interactions without introducing steric hindrance.

The detection time from sample application to the generation of an electrochemical signal is less than 15 min, offering a significant advantage over ELISA (~2 h) and PCR (1.5–3 h), considering also the sample preparation requirements. A comparison of time, sensitivity, and field applicability with reference methods is summarized in Table 3.

We also conducted a literature review comparing the key analytical parameters (LOD, linear range, and response time) of our biosensor with existing diagnostic methods. These data are summarized in Table 4. The results demonstrate that our biosensor achieves competitive detection limits and a significantly shorter response time while retaining a straightforward design suitable for future field deployment.

It is important to note that this work represents an initial stage of research. Further studies are necessary to evaluate the long-term stability, sterility, reproducibility, and performance of the biosensor in real-world biological or environmental samples. Future work will also include selectivity testing against possible interferents such as other *Bacillus* species and improvement of storage conditions and sensor reusability.

Despite these limitations, the results indicate the promising potential of the proposed biosensor as a rapid, sensitive, and selective tool for the detection of *B. anthracis* spores. This approach may be particularly valuable in field diagnostics and biosafety-related applications.

## 5. Conclusions

The main achievement of this study is the development of a methodology for modifying electrochemical transducers, enabling the selective and sensitive detection of *Bacillus anthracis* spores, even in the presence of other bacteria, including closely related *Bacillus* species. The obtained biosensors, based on modified gold electrodes, demonstrated high sensitivity and specificity towards the target antigen and, when used in a laboratory cyclic voltammetry system, allowed for the detection of *B. anthracis* spores at concentrations as low as 10^3^ CFU/mL. The measurement process is performed in a single step without the need for complex sample preparation, requiring only a microliter-scale sample volume. Furthermore, the use of disposable electrodes eliminates the risk of cross-contamination and the need for decontamination after measurement.

In comparison to standard diagnostic techniques such as PCR and ELISA, the developed biosensor significantly reduces analysis time. While PCR remains the most sensitive method, its infrastructure requirements and lengthy processing make it unsuitable for field applications. ELISA, although simpler, still requires long assay times and is less amenable to miniaturization. By contrast, the proposed electrochemical method is rapid, portable, and user-friendly, offering clear advantages for on-site detection of biological threats.

The high miniaturization potential of electrochemical sensing systems, combined with low operational requirements, makes this approach well-suited for integration into portable analyzers designed for field diagnostics. The biosensor platform is also highly versatile and can be adapted to detecting other microorganisms by modifying the recognition layer with appropriate antibodies. Tests conducted in both model laboratory conditions and using live cells and spores confirmed the applicability of this solution for environmental samples, including swabs, air, and solid materials after suspension and particle removal.

Although the current system provides qualitative detection of *B. anthracis* without quantification or discrimination between vegetative and spore forms, the results highlight its significant potential for field analysis at the scene of potential contamination. Future work will focus on integrating the biosensor with a miniaturized electrochemical reader to enable fully portable detection, as well as validating its performance in real environmental matrices such as soil, air, and water.

While the biosensor shows a promising detection capability, it is still in the early development stage. Future work must include cross-reactivity testing against a broader panel of pathogens, long-term stability assessment, sterility assurance, and validation using real environmental samples.

## Figures and Tables

**Figure 1 sensors-25-05948-f001:**
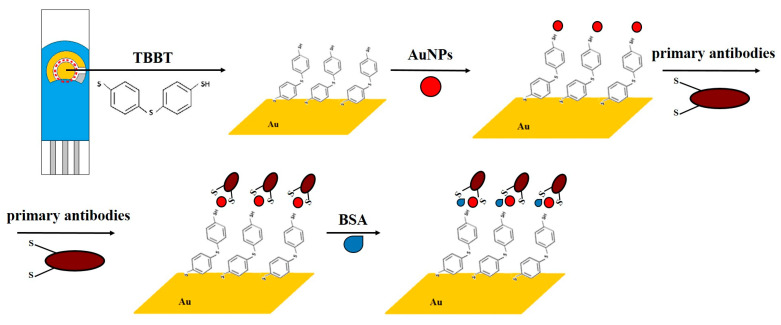
Modification of gold electrode: schematic illustration of the stages of modification. Cleaning of the gold electrode surface, formation of a self-assembled monolayer using 4-tert-butylbenzenethiol, deposition of gold nanoparticles, covalent immobilization of primary monoclonal antibodies specific to *Bacillus anthracis*, and passivation with bovine serum albumin to prevent nonspecific adsorption.

**Figure 2 sensors-25-05948-f002:**
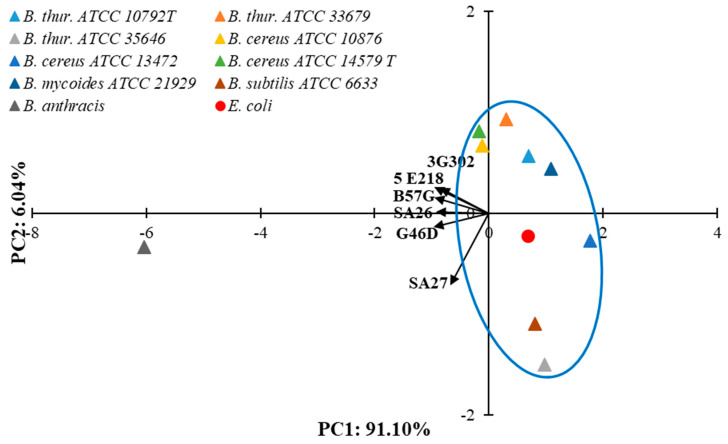
The Principal Component Analysis (PCA) biplot describing the specificity of the tested monoclonal antibodies against *Bacillus* spores and the differences between antibodies.

**Figure 3 sensors-25-05948-f003:**
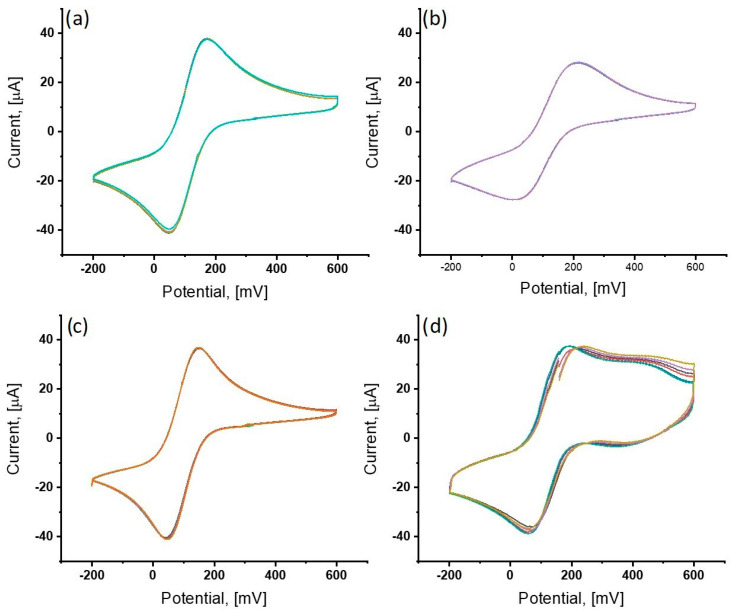
Cyclic voltammograms recorded for individual stages of modification of the working electrode surface: (**a**) electrochemical cleaning of the electrode; (**b**) working electrode with a layer of TBBT; (**c**) working electrode with a reconstituted layer of gold nanoparticles; (**d**) working electrode with deposited primary antibodies SA27.

**Figure 4 sensors-25-05948-f004:**
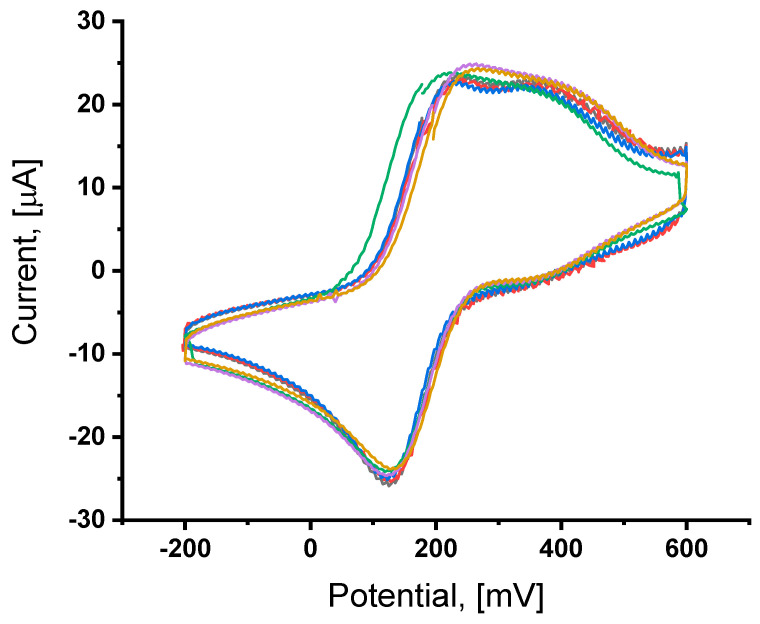
Cyclic voltammograms after incubation with *B. anthracis*. The measurement was performed using a concentration of 10^8^ CFU/mL of *B. anthracis spores* in PBS. Five CV cycles were conducted for each electrode.

**Figure 5 sensors-25-05948-f005:**
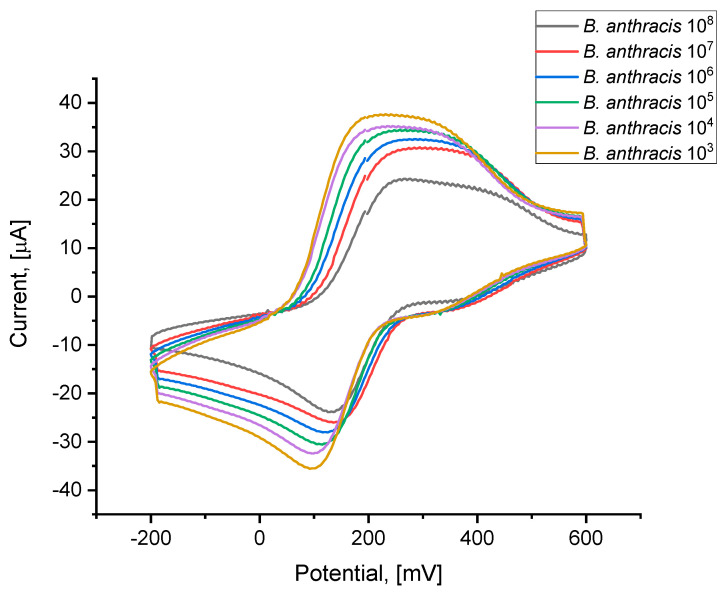
A graph showing the change in the recorded voltammogram depending on the concentration of *B. anthracis* from 10^3^ to 10^8^ CFU/mL in 0.1 M PBS at pH 7.4 and 1 mM of ferrocyanide.

**Figure 6 sensors-25-05948-f006:**
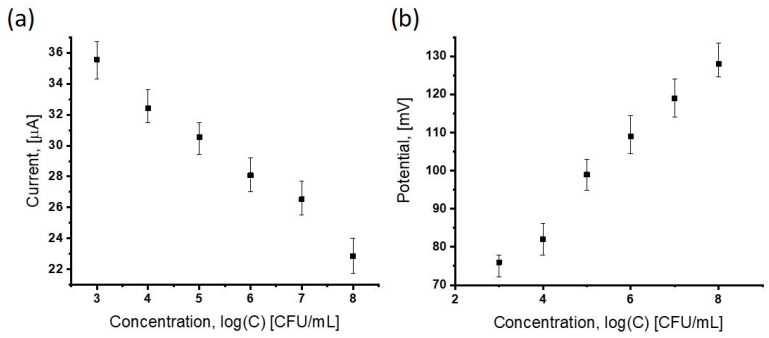
A plot showing the relationship between voltage and current changes as a function of *B. anthracis* concentration from 10^3^ to 10^8^ CFU/mL in 0.1 M PBS and 1 mM of ferrocyanide at pH 7.4: current vs. concentration (**a**), potential vs. concentration (**b**). Each point corresponds to the current and potential values at the maximum of the reduction peak.

**Figure 7 sensors-25-05948-f007:**
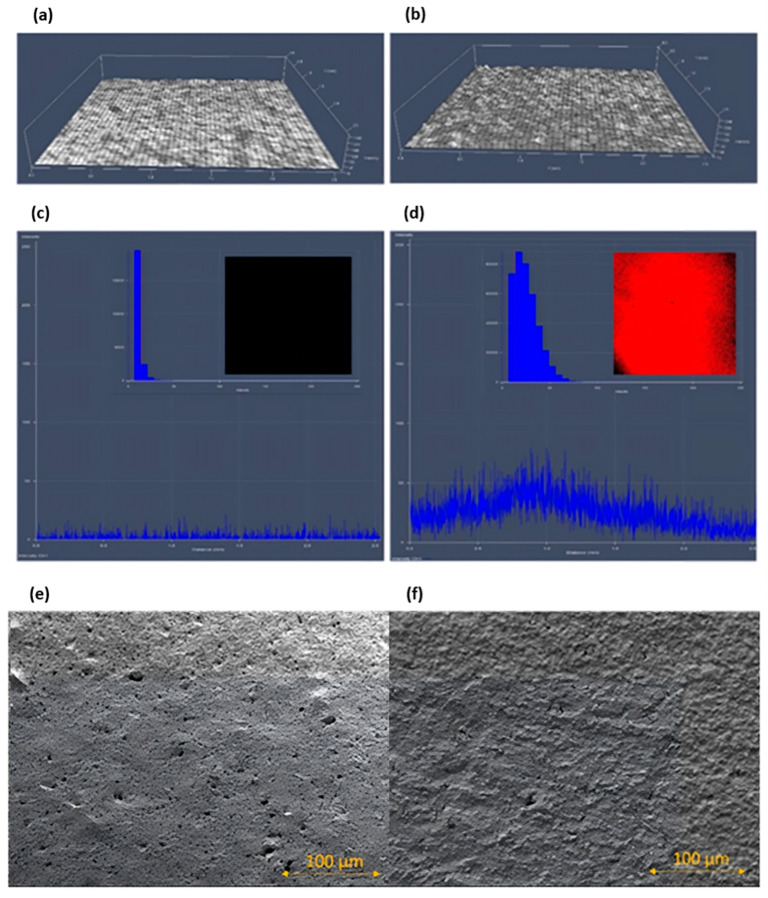
The scan of the electrode surface obtained using the CSLM (**a**) without antibodies, (**b**) coated with primary and secondary antibodies. The fluorescent image from the CSLM (**c**) without antibodies, (**d**) coated with primary and secondary antibodies. Image obtained using the STEM, magnification (1000×), (**e**) without antibodies, (**f**) coated with primary and secondary antibodies.

**Table 1 sensors-25-05948-t001:** Results of antibody selectivity tests.

Bacteria	% of Spore Binding by Antibodies					
	SA26	G46D	B57G	5 E218	3G302	SA27
*B. thur.* ATCC 10792T	25.9	18.7	21.9	28.0	51.7	19.7
*B. thur.* ATCC 33679	24.3	23.2	29.3	40.3	51.7	15.1
*B. thur.* ATCC 35646	23.8	27.6	20.8	13.0	12.5	57.0
*B. cereus* ATCC 10876	42.7	19.9	48.2	44.6	22.7	20.7
*B. cereus* ATCC 13472	22.8	15.4	13.8	16.5	10.4	24.1
*B. cereus* ATCC 14579 T	23.4	34.2	47.2	39.8	43.1	17.3
*B. mycoides* ATCC 21929	19.0	22.1	25.0	24.4	34.2	16.0
*B. subtilis* ATCC 6633	12.0	21.6	23.0	23.0	32.0	57.6
*B. anthracis*	85.6	86.3	87.4	88.7	98.8	86.5
*E. coli*	33.3	28.7	32.0	22.6	13.9	28.1

**Table 2 sensors-25-05948-t002:** Selectivity coefficients S for individual antibodies.

Antibody	SA26	G46D	B57G	5 E218	3G302	SA27
Selectivity coefficient S [%]	44.1	47.2	38.1	38.63	38.25	38.02

**Table 3 sensors-25-05948-t003:** A comparison of the proposed biosensor with reference methods.

Method	Total Analysis Time	Sample Preparation	LOD	Field Applicability
Proposed biosensor	<15 min	Minimal	10^3^ CFU/mL (estimated)	High
PCR	90–180 min	DNA extraction required	10–100 CFU/mL	Low
ELISA	~120 min	Multiple steps	10^3^–10^4^ CFU/mL	Moderate

**Table 4 sensors-25-05948-t004:** Analytical parameters of *Bacillus anthracis* detection methods based on the literature.

Method/Reference	LOD	Linear Range	Response Time
This work	10^3^ CFU/mL	10^4^–10^7^ CFU/mL	<15 min
PCR [33]	10 CFU/mL	10–10^6^ CFU/mL	1.5–3 h
ELISA [34]	10^3^ CFU/mL	10^3^–10^6^ CFU/mL	~2 h
LFA [35]	~10^6^ spores/mL	-	Rapid (min–h)
FET-based biosensor [36]	10^2^ CFU/mL	10^2^–10^5^ CFU/mL	~30 min
SPR biosensor [37]	10^2^–10^3^ CFU/mL	10^2^–10^6^ CFU/mL	~45 min

## Data Availability

The original contributions presented in this study are included in the article/Appendix A. Further inquiries can be directed to the corresponding author(s).

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
