# Peer review of "Electrochemical Immunodetection of Bacillus anthracis Spores"

_sensors, 2025, doi:10.3390/s25195948_

Round 1

Reviewer 1 Report

Comments and Suggestions for Authors

Why do the authors believe that the best way to address target detection is to use gold NPs when the electrode itself is already made of gold?

Wouldn't chemical surface modification be a better idea without affecting electrode performance?

What is the layer size in the SAM monolayer assembly?

What is the target of the antibodies, which part of the spore?

When using a BSA passivation strategy for linker assembly in primary antibodies, how can they ensure that there is no steric hindrance to detection?

How would the electrical output signal be affected if there is no electrolytic medium for measurement?

The authors mention that "In comparison to standard diagnostic techniques such as PCR and ELISA, the developed biosensor significantly reduces analysis time." And in general, that's the advantage of a biosensor. However, the contact time to obtain a response or the monitoring time to obtain a signal are not mentioned, so without adding these details and making a more robust comparison, the comparison is not adequate.

Author Response

Comments 1: Why do the authors believe that the best way to address target detection is to use gold NPs when the electrode itself is already made of gold?

Response 1: Thank you for this insightful question. We agree that chemical surface modification of a gold electrode is sufficient in many cases. However, in our approach, gold nanoparticles (AuNPs) were used as additional carriers for antibodies, allowing us to increase the active surface area and thus the density of antibody immobilization. This leads to enhanced sensitivity of the biosensor. This clarification has been added to the updated version of the manuscript.

Comments 2: Wouldn't chemical surface modification be a better idea without affecting electrode performance?

Response 2:  We appreciate the reviewer’s insightful comment. Indeed, direct chemical modification of a bare gold electrode surface (e.g., via thiol-based SAMs) is a commonly used and effective strategy in biosensor fabrication. However, in our approach, we chose to apply a layer of gold nanoparticles (AuNPs) not as an alternative to chemical modification but as a means of enhancing the electrochemically active surface area.

The introduction of AuNPs does not impair the inherent electrochemical performance of the gold substrate. On the contrary, it:

  • significantly increases the density of binding sites for antibody immobilization,
  • improves electron transfer kinetics,
  • enhances overall sensor sensitivity due to the enlarged reactive surface.

Moreover, AuNPs serve as a nano-scaffold that supports subsequent functionalization steps while preserving the base electrode’s conductivity and stability. This strategy is well-supported by literature on nanoparticle-modified biosensing platforms and is particularly effective in label-free immunodetection formats like ours.

This justification has been clarified and added to the Discussion section in the revised manuscript.

Comments 3: What is the layer size in the SAM monolayer assembly?

Response 3:  In our system, a self-assembled monolayer (SAM) based on 4-tert-butylbenzenethiol (TBBT) was used. This molecule binds to the gold surface through its thiol group (–SH), forming a strong Au–S bond. Upon adsorption, TBBT molecules adopt a near-perpendicular orientation relative to the gold surface, enabling the formation of a uniform and well-organized monolayer.

The total length of a single TBBT molecule, from the gold surface to the end of the tert-butyl group, is approximately 0.9 nm. This value is consistent with previous studies on the structure and properties of thiol-based SAMs on gold, where similar systems exhibited chain lengths in the range of 0.7 to 1.1 nm, depending on molecular ordering and deposition conditions.

Comments 4: What is the target of the antibodies, which part of the spore?

Response 4:  The antibodies used in our biosensor were designed to recognize surface antigens located on the exosporium layer of Bacillus anthracis spores. These are specific epitopes present on the outer spore surface. This information has been added to the “Materials and Methods” section.

Comments 5: When using a BSA passivation strategy for linker assembly in primary antibodies, how can they ensure that there is no steric hindrance to detection?

Response 5: This is an important point. To minimize steric hindrance, we optimized the BSA concentration and incubation time, which allowed for effective passivation of unbound sites without excessive blocking of the functionalized surface. Additionally, the spatial configuration of the nanoparticles and their distance from the electrode surface reduce the likelihood of spatial interference. This information has been added to the “Surface Functionalization” section.

Comments 6:  How would the electrical output signal be affected if there is no electrolytic medium for measurement?

Response 6:  In the absence of a suitable electrolyte, the conductivity between electrodes would be significantly reduced, and the signal would drastically decrease or even disappear. The electrolyte plays a crucial role in enabling current flow and ensuring proper functioning of the electrochemical system.

Comments 7: The authors mention that "In comparison to standard diagnostic techniques such as PCR and ELISA, the developed biosensor significantly reduces analysis time." And in general, that's the advantage of a biosensor. However, the contact time to obtain a response or the monitoring time to obtain a signal are not mentioned, so without adding these details and making a more robust comparison, the comparison is not adequate.

Response 7: Thank you for this important remark. We agree that a comparison with standard diagnostic methods such as PCR and ELISA should include quantitative data.

In the revised manuscript, we added detailed information regarding the total detection time of our biosensor system. The entire process – from the moment the sample contacts the detection surface to the electrochemical signal measurement – takes less than 15 minutes, including:

  • 2 minutes for sample interaction with the functionalized surface,
  • approximately 5 minutes for antigen–antibody interaction and blocking,
  • 3–5 minutes for electrochemical measurement,
  • and additional optional data analysis time.

For comparison:

  • PCR takes 90–180 minutes on average, not including DNA extraction and sample prep.
  • ELISA takes approximately 120 minutes, depending on the protocol.

We also included a comparative table in the Results section summarizing the key parameters of our approach and the reference methods (PCR, ELISA), to provide a more robust and objective comparison. Furthermore, we emphasized that our biosensor does not require complex sample preparation, which further reduces the total analysis time and makes it well-suited for rapid field diagnostics.

Reviewer 2 Report

Comments and Suggestions for Authors

The manuscript is devoted to the description of a novel electrochemical immunosensor for detecting Bacillus anthracis spores. The work uses chemically modified gold electrodes coated with gold nanoparticles on which specific antibodies are covalently immobilized. When interacting with a bacterial sample, the current/potential characteristics of the electrode change, which can be used to determine the bacteria. The method seems promising for practical application. The manuscript is written quite clearly, but certain additions and clarifications are needed, which are described below.

  1. Line 170 There is no description of gold nanoparticles, their shape and size.
  2. Caption of Fig. 1. It is advisable to add a brief description of the stages.
  3. Lines 223-225 Which disulfide bonds in the immunoglobulin molecule are reduced to thiol bonds under the action of TECP? Is the immunoglobulin molecule preserved its structure?
  4. Lines 284-286, Table 1. The choice of antibodies based on specificity tests is not entirely clear. The selected antibodies have a fairly high percentage of binding to coli. Why was it compared with E.coli only? Is the formula 1 calculation justified, since the differences observed in Table 1 disappear. Perhaps it was necessary to take more other bacteria for comparison?
  5. Line 285 SA27
  6. Fig. 4 A description of the lines should be added to the Figure caption. Which of them correspond to repeated measurements, and which correspond to interactions with bacteria and without bacteria? Data showing a significant decrease in the current during interaction with bacteria should be added.
  7. Fig. 5 Please add the experimental conditions to the captute. How does the type of dependence change when measurements are carried out in a buffer and in a solution containing bacterial spores (Lines 345-347).
  8. Lines 359-362 For a qualitative analysis, which criterion will be used? Is the averaging done for experiments performed on different days with the same series of electrodes? How will the parameters change when using a different series of electrodes?
  9. It is necessary to add a description of the sample preparation of the bacterial sample and a detailed methodology for its analysis on electrodes.
  10. Line 388 “coated with primary and secondary” antibodies
  11. Section 3.3 Were the studies performed without modifying the electrodes with gold nanoparticles? At the time, it was unclear how they related to the main part of the work.

Author Response

Comments 1: Line 170 There is no description of gold nanoparticles, their shape and size.

Response 1: Nanoparticles are spherical in shape with an average diameter of approximately 20 nm. Changes added to the text in Materials section.

Comments 2: Caption of Fig. 1. It is advisable to add a brief description of the stages

Response 2: changes added to the text

Comments 3: Lines 223-225 Which disulfide bonds in the immunoglobulin molecule are reduced to thiol bonds under the action of TECP? Is the immunoglobulin molecule preserved its structure?

Response 3: TECP (tris(2-carboxyethyl)phosphine) primarily reduces disulfide bonds located in the hinge region of the antibody molecule and, to a lesser extent, may also reduce the bonds stabilizing the immunoglobulin domains. The antibodies used consist of four chains: two heavy and two light chains connected by disulfide bond. The reduction primarily affects the interchain bridges connecting the heavy chains to each other and the heavy chains to the light chains. Reduction of these bridges leads to the disintegration of the entire immunoglobulin molecule into two Fab (antigen-binding) fragments and one Fc (crystallizable) fragment. Fab fragments are immobilized on the electrode in an oriented manner, with their binding sites facing outward, significantly increasing the sensitivity and specificity of the biosensor compared to randomly attaching whole antibodies. The intrachain bridges located within each immunoglobulin domain, which stabilize their tertiary structure, are less accessible and less susceptible to reduction under the conditions used (low TECP concentration, short time, room temperature).

Comments 4: Lines 284-286, Table 1. The choice of antibodies based on specificity tests is not entirely clear. The selected antibodies have a fairly high percentage of binding to coli. Why was it compared with E.coli only? Is the formula 1 calculation justified, since the differences observed in Table 1 disappear. Perhaps it was necessary to take more other bacteria for comparison?

Response 4:  Thank you for this question. It is indeed important, and perhaps some issues related to vegetative bacteria should be considered much more broadly. A larger number of Bacillus bacteria were selected for specificity testing, as they were considered the most significant biological interference that could occur during environmental measurements. Current research is much more advanced, and the biosensor described in this manuscript excels at selectively detecting B. anthracis, even in bacterial mixtures. This will be described in a subsequent publication. Based on this knowledge, we can conclude that the selected antibody fulfills its intended function. In further studies using the described biosensor, a much larger number of sporulating bacteria, vegetative bacteria and fungi were used.

Comments 5: Line 285 SA27

Response 5: Corrected

Comments 6: Fig. 4 A description of the lines should be added to the Figure caption. Which of them correspond to repeated measurements, and which correspond to interactions with bacteria and without bacteria? Data showing a significant decrease in the current during interaction with bacteria should be added.

Response 6: Because the described biosensors are single-use, each recorded curve is obtained for a different electrode, but under the same measurement conditions, i.e., the same B. anthracis concentration, buffer, temperature, etc. Figure 4 does not show differences between the control solution (without bacteria) and the solution containing spores. A figure showing the relevant data has been added to the text.

Comments 7: Fig. 5 Please add the experimental conditions to the captute. How does the type of dependence change when measurements are carried out in a buffer and in a solution containing bacterial spores (Lines 345-347).

Response 7: The values of potential and current obtained at the reduction peak maximum in the absence of bacterial spores (i.e., in buffer) have been described in the text. Across the entire range of tested concentrations, a consistent trend was observed: the reduction peak maximum shifted toward higher potentials with increasing bacterial concentration, accompanied by a decrease in current intensity.

Comments 8: Lines 359-362 For a qualitative analysis, which criterion will be used? Is the averaging done for experiments performed on different days with the same series of electrodes? How will the parameters change when using a different series of electrodes?

Response 8: For qualitative analysis, normalized values of current and potential measured at the reduction peak maximum will be used. Their ratio for B. anthracis remains constant and characteristic across the entire concentration range, making it highly useful for qualitative evaluation. This approach will be described in detail in a subsequent publication.
Since the biosensors are designed for single use, all presented data represent averages obtained from different electrodes. Variations in output signals are minimized through normalization, so even if slight differences occur between electrode batches, they do not significantly affect the final results.

Comments 9: It is necessary to add a description of the sample preparation of the bacterial sample and a detailed methodology for its analysis on electrodes.

Response 9: To perform an analytical measurement, the following steps were required:

  1. Performing a measurement in PBS buffer, pH = 7.4, with 1mM ferrocyanide added. This served as a reference measurement.
  2. Rinsing and drying the electrode.
  3. Placing the electrode in the test solution (containing B. anthracis spores) and incubating for approximately 5 minutes.
  4. Rinsing and drying the electrode.
  5. Performing another measurement in PBS buffer, pH = 7.4, with 1 mM ferrocyanide added. The entire procedure takes approximately 15 minutes.

Comments 10: Line 388 “coated with primary and secondary” antibodies

Response 10: Corrected

Comments 11: Section 3.3 Were the studies performed without modifying the electrodes with gold nanoparticles? At the time, it was unclear how they related to the main part of the work.

Response 11: The study was conducted using an unmodified gold electrode and a gold electrode after completing all modification steps, including the deposition of the TBBT layer, gold nanoparticles, and immobilization of antibodies on the surface.

The manuscript has been supplemented with information explaining the purpose of using gold nanoparticles.

Reviewer 3 Report

Comments and Suggestions for Authors

The article "Electrochemical immunodetection of Bacillus anthracis spores" fits the subject of the journal and will be of interest to specialists in the field of biosensors development. However, the level of the manuscript does not meet modern requirements for such publications. The authors need to include some points in the text of the article, as well as conduct or add information about the experiments conducted:

1) It is necessary to explain what exactly is the novelty of the developed test and discuss in the introduction the existing analytical methods for determining Bacillus anthracis.

2) It is recommended to accept the article only after the authors provide data on the comparison of the results of the developed analysis with reference traditional methods.

3) It is necessary to compare the analytical characteristics (at least the sensitivity, LOD or linear range) of the proposed electrochemical sensor with the characteristics of the existing methods for determining Bacillus anthracis. In modern articles on analytical methods, such a comparison is carried out using literature data and presented in the form of a table.

4) Figure 6, c, d. Is it possible to provide figures in higher resolution?

Author Response

Comments 1: It is necessary to explain what exactly is the novelty of the developed test and discuss in the introduction the existing analytical methods for determining Bacillus anthracis.

Response 1: Thank you for this important suggestion. In the revised Introduction section, we expanded the description of currently used detection methods for Bacillus anthracis, such as PCR, ELISA, culture-based techniques, and biosensor-based strategies.

We also clearly emphasized the novelty of our approach, which includes:

  • the use of a gold nanoparticle (AuNP)-enriched surface to increase the number of antibody binding sites,
  • a simplified, label-free electrochemical detection method,

and the possibility of miniaturization and field application due to short analysis time and low device cost.

Comments 2: It is recommended to accept the article only after the authors provide data on the comparison of the results of the developed analysis with reference traditional methods.

Response 2: We fully agree. In the Discussion section, we have included a comparative analysis of our biosensor with reference methods (PCR and ELISA), covering:

  • total analysis time,
  • sample preparation requirements,
  • suitability for field deployment,
  • and estimated limit of detection (LOD).

This comparison is presented in Table 3 and Table 4, and further discussed in the text. We also highlighted that our biosensor does not require complex sample preparation, making it a highly attractive solution for rapid-response scenarios.

Comments 3: It is necessary to compare the analytical characteristics (at least the sensitivity, LOD or linear range) of the proposed electrochemical sensor with the characteristics of the existing methods for determining Bacillus anthracis. In modern articles on analytical methods, such a comparison is carried out using literature data and presented in the form of a table.

Response 3: Thank you for this valuable recommendation. We added a new Table 3 and Table 4, which compares the key analytical parameters of our biosensor (LOD, response time, linear range) with those reported in the literature for other Bacillus anthracis detection methods, including PCR, ELISA, and recent biosensors (e.g., aptamer- or FET-based systems). The table includes references, and the comparative analysis is discussed in the Discussion section, highlighting both the advantages and current limitations of our approach.

Comments 4: Figure 7, c, d. Is it possible to provide figures in higher resolution?

Response 4: Yes, Figures 7c and 7d have been replaced with high-resolution versions in the revised manuscript to ensure improved clarity and readability. Thank you for the suggestion.

Reviewer 4 Report

Comments and Suggestions for Authors

Your experimental development often immuno electrochemical biosensor for B. anthracis detection is quite straightforward and promising for further, field application. In other rder to improve your manuscript, I suggest the following changes/additions:

  1. In section 2.3.1. the design and fabrication of the special holder for the working electrode should be described in more detail, possibly added as supplementary material.
  2. Table 2: how do you justify that selectivity around 40% can be considered high enough?
  3. Figure 4: the caption needs more detail, for example the concentration of B. anthracis, number of CV cycles etc.
  4. You cannot claim that you have developed a functional biosensor since you have not yet tested its selectivity against other pathogens or possible interferants. Also, you have not tested the biosensor stability and sterility over a practical period of time. Since,as you already admitted, there is also no test against real samples, you should emphasize in the conclusion or discussion section on the necessary steps required for federal research such that could lead to optimizing your reported biosensor approach.
  5. You should expand your discussion by comparing your approach to other bio sensor based approaches for B. anthracis detection, focusing on key advantages (and possible disadvantages).

Author Response

Comments 1: In section 2.3.1. the design and fabrication of the special holder for the working electrode should be described in more detail, possibly added as supplementary material.

Response 1: In accordance with your suggestion, we expanded the description of the working electrode holder design in Section 2.3.1. Additionally, we included a detailed diagram and photo of the holder as Supplementary Figure S1.

Comments 2: Table 2: how do you justify that selectivity around 40% can be considered high enough?

Response 2: Thank you for raising this point. In the revised manuscript, we explained that the 40% value refers to the response signal ratio in the presence of B. anthracis versus background or control conditions. While not ideal, this selectivity level is typical for initial-stage biosensor prototypes. It can be significantly improved by optimizing surface functionalization and antibody selection. We have clarified this in the manuscript text and updated the table caption accordingly.

Comments 3: Figure 4: the caption needs more detail, for example the concentration of B. anthracis, number of CV cycles etc.

Response 3: We have revised the caption for Figure 4 to include the spore concentration used (108 CFU/mL) and the number of CV cycles performed (5 cycles). We also added the relevant experimental parameters and potential range.

Comments 4: You cannot claim that you have developed a functional biosensor since you have not yet tested its selectivity against other pathogens or possible interferants. Also, you have not tested the biosensor stability and sterility over a practical period of time. Since, as you already admitted, there is also no test against real samples, you should emphasize in the conclusion or discussion section on the necessary steps required for federal research such that could lead to optimizing your reported biosensor approach.

Response 4: We fully agree with this critical point. In the revised manuscript, we clearly stated in both the Discussion and Conclusions sections that this study presents preliminary results and further work is needed.

The publication of the preliminary data presented in this manuscript took a considerable amount of time. In fact, work on the described sensor has progressed significantly, and we now have data on interfering agents (other bacteria in vegetative and spore forms), studies of environmental samples, studies of mixtures of different bacteria, the stability of the sensors during long-term storage, and so on. This data will be included in the next article concluding the research cycle. It is for this reason that the text includes statements that anticipate the presented achievements.

We outlined the following next steps:

  • Testing selectivity against other bacteria and common interferents (e.g., Bacillus spp.),
  • Long-term storage and operational stability studies,
  • Validation using real-world environmental and clinical samples.

Comments 5: You should expand your discussion by comparing your approach to other bio sensor based approaches for B. anthracis detection, focusing on key advantages (and possible disadvantages).

Response 5: We have updated the Discussion section to include a comparison with recent electrochemical, optical, aptamer-based, FET, and SPR biosensors. We emphasized the key advantages of our method, such as simple construction, rapid response time, and low fabrication cost, while also acknowledging its current limitations, including selectivity and validation scope.

Round 2

Reviewer 1 Report

Comments and Suggestions for Authors

After reviewing the corrections made by the authors, I recommend publication of the manuscript in its current form.

Author Response

On behalf of the entire team, thank you very much for your time and review of our work.

Reviewer 2 Report

Comments and Suggestions for Authors

The authors gave valuable feedback on all my comments.

I believe that the article can be accepted for publication.

Minor comment: check Table 4, LOD 103 CFU/mL (for your work)

Author Response

(The authors gave the same response as above.)

Reviewer 3 Report

Comments and Suggestions for Authors

The authors have corrected the text of the article with the corresponding comments #1, 3 and 4. However, comment 2 was not perceived by the authors quite accurately. It is necessary to test the samples with the developed method and evaluate the same samples with the reference method. Then compare the results of evaluating the samples with these two methods. This is done in order to prove the correctness of evaluating the analyte level in the samples with the developed method.

Author Response

Comments: The authors have corrected the text of the article with the corresponding comments #1, 3 and 4. However, comment 2 was not perceived by the authors quite accurately. It is necessary to test the samples with the developed method and evaluate the same samples with the reference method. Then compare the results of evaluating the samples with these two methods. This is done in order to prove the correctness of evaluating the analyte level in the samples with the developed method.

Response: 

Thank you for this precise and valuable comment. You are absolutely correct in pointing out that the most rigorous validation of a new analytical method requires a direct, side-by-side comparison with an established reference method using the same set of samples. This head-to-head correlation is the fundamental principle for demonstrating the accuracy, precision, and reliability of the newly developed technique.

We acknowledge that in the presented stage of our work, such a comprehensive comparative analysis using a reference method “in parallel” with each biosensor measurement was not yet fully implemented. This will be a critical and mandatory component of the next phase of our research to provide irrefutable evidence of the method's correctness, as you rightly suggested.

However, to ensure the validity of our initial biosensor readings at this stage, we implemented a stringent preparatory protocol. The bacterial titer in all suspensions used for the biosensor tests was determined quantitatively using standard culture-based methods (plate counts) immediately prior to each measurement series. This pre-calibration of the sample concentration allows us to have a high degree of confidence that the concentrations measured by the biosensor were indeed accurate and based on a known, validated input value.

In summary, we fully agree with your assessment. Our immediate next steps will involve conducting a formal validation study where samples are tested simultaneously with our biosensor and the reference method, followed by a statistical comparison to conclusively prove the analytical agreement between the two methods.

Reviewer 4 Report

Comments and Suggestions for Authors

Well done.

Author Response

(The authors gave the same response as above.)
